# Heterogeneous Domain Generalization for Single-Source Cross-Dataset Person ReID: An Adaptive Adversarial Augmentation Approach

## Abstract

Despite the significant advances in supervised person re-identification (ReID) methods, these models exhibit performance degradation in unseen domains. Domain generalization (DG) is applied to alleviate this issue, but most existing DG methods assume consistent class spaces between source and target domains. We propose Adaptive Adversarial Augmentation (AAA), a Heterogeneous Domain Generalization (HDG) approach tailored for single-source cross-dataset ReID. AAA jointly trains a feature extractor alongside a Domain Adversarial Network (DAN) and a Class Adversarial Network (CAN) to enhance the feature extractor's robustness to both domain shifts and class space changes. Additionally, we propose a diversity-based perturbation impact factor, dynamically tuning the perturbation influence aligned with the diversity of learned embeddings, thus providing a flexible augmentation strategy. Experimental results demonstrate that our method surpasses state-of-the-art methods on large-scale cross-dataset ReID benchmarks.

## 1 Introduction

Person re-identification (ReID) aims to match persons of the same identity across non-overlapping cameras under various viewpoints and locations. Notable advancements have been achieved in the supervised setting, where training (source domain) and testing sets (target domain) are distinct partitions of the same dataset (Zhou et al., 2020a; Zhang et al., 2020; Chen et al., 2020; Liu et al., 2020). However, this supervised paradigm is hardly applicable since it suffers significant performance deterioration on unseen target domains (cross-dataset ReID) due to domain shifts (Pan & Yang, 2009). As shown in Figure 1, compared to the supervised ReID, the cross-dataset ReID is more challenging since the target domains have distinct domain distribution and class spaces. Unsupervised Domain Adaptation (UDA) methods (Liu et al., 2019; Zhai et al., 2020; Wang et al., 2020) have been proposed for cross-dataset ReID by training models on labelled source domains and adapting the model to unlabelled target domains. Although UDA presents a more feasible approach than the supervised one, it still requires data collection and model adaptation for each new target domain. Domain Gen-

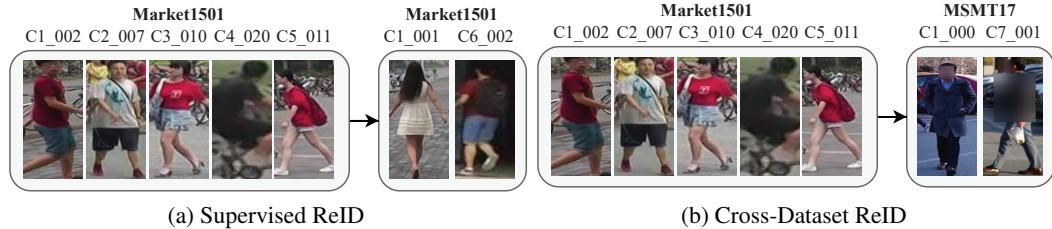

Figure 1: In the supervised setting, target domains derive from the same dataset as source domains, potentially sharing domain and identity spaces. Conversely, in the cross-dataset setting, target domains are drawn from distinct datasets, leading to entirely disjoint domain and identity spaces.

eralization (DG) offers a solution without the need for target domain data (Wang et al., 2022; Zhou et al., 2022). Nonetheless, most DG approaches assume consistent class space across the source and target domains (Muandet et al., 2013; Sun et al., 2022; Arpit et al., 2022), limiting their applicability in the cross-dataset ReID problem, where the class spaces are inherently disjoint. The seminal work of Li et al. (2019) paves the way by introducing the Heterogeneous Domain Generalization (HDG), embracing both domain shifts and disjoint class spaces, and Zhou et al. (2020b) extended HDG to cross-dataset ReID. However, existing HDG methods primarily focus on meta-learning (Zhao et al., 2021; Choi et al., 2021) or domain invariant representations (Song et al., 2019; Jin et al., 2020; Zhou et al., 2021a), which face the challenge of overfitting source domain data since the source domain data remains unchanged (Zhou et al., 2020b;c; 2022).

This paper proposes Adaptive Adversarial Augmentation (AAA), a Heterogeneous Domain Generalization approach for single-source cross-dataset ReID, without access to target domain data. Single-source cross-dataset ReID represents the most challenging setting in ReID tasks, aiming to train a model on one source dataset to generalize across multiple unseen target datasets. Existing works focus on improving the feature extractor's robustness to domain shifts by augmenting source domain data with perturbations generated from a concurrently trained Domain Adversarial Network (DAN) (Shankar et al., 2018; Zhou et al., 2020b). Beyond that, we propose a Class Adversarial Network (CAN), jointly trained with the feature extractor and DAN, to enhance the feature extractor's robustness to the changes in class space. In contrast to existing methods utilizing a static perturbation impact factor (Shankar et al., 2018; Zhou et al., 2020b; Sun et al., 2022), we propose a diversity-based perturbation impact factor that dynamically modulates the perturbation influence according to the diversity of learned embeddings, thereby offering a flexible augmentation strategy.

To evaluate the effectiveness of AAA, we conduct experiments on three benchmark datasets for single-source cross-dataset person re-identification: CUHK03 (Li et al., 2014), Market1501 (Zheng et al., 2015), and MSMT17 (Wei et al., 2018). These datasets encompass a range of ReID tasks and include multiple camera domains. The experimental results demonstrate that AAA surpasses the state-of-the-art DG and HDG methods on the evaluated datasets. We perform ablation studies to demonstrate the effectiveness of each component within AAA. Furthermore, we visualize the generated images and feature embeddings, offering insights into the underlying mechanisms contributing to the proposed approach. The code is available in https://anonymous.4open.science/r/HDG-776E/.

## 2 RELATED WORK

**Cross-Dataset Person Re-Identification.** Significant progress has been made in fully-supervised person ReID in the past decade, particularly with deep learning-based approaches (Qian et al., 2018; Zhang et al., 2019; Tay et al., 2019; Zhou et al., 2019; Zhong et al., 2020). These methods demonstrate impressive performance when the training and test sets share similar distributions. However, they generalize poorly to previously unseen domains (datasets) due to distribution shifts across domains. When the target domain data is accessible, even without annotations, it can be leveraged through Unsupervised Domain Adaptation (UDA) techniques. UDA-based ReID methods can be broadly categorized into three groups: style transfer (Chen et al., 2019; Liu et al., 2019), attribute recognition (Qi et al., 2019), and target-domain pseudo-label estimation (Zhong et al., 2019; Wang & Zhang, 2020). While the UDA methods are effective for cross-dataset person ReID, they still require data collection and model adaptation for each new target domain. In practice, acquiring target domain data can often prove impractical, underscoring the need for domain generalization techniques. For example, obtaining data encompassing all identities under various conditions, such as camera brands, camera views, and weather conditions, is infeasible (Yue et al., 2019).

**Domain Generalization.** Domain Generalization (DG) was first introduced by Blanchard et al. (2011) and later formalized by Muandet et al. (2013). DG aims to train models capable of generalizing to unseen domains without requiring target domain data (Zhou et al., 2022; Wang et al., 2022). However, most existing DG methods focus on closed-set tasks (Muandet et al., 2013; Li et al., 2018; Qiao et al., 2020), assuming consistent class spaces across source and target domains, which conflicts with the disjoint class space case in cross-dataset ReID. Li et al. (2019) pioneered addressing this gap by introducing the Heterogeneous Domain Generalization (HDG) paradigm, which accommodates both domain shifts and disjoint class spaces. Yet, their meta-learning-based approach for HDG is framed within zero-shot learning, limiting its applicability to the ReID challenge. Subse-

quently, Zhou et al. (2020b) explored a data augmentation strategy for HDG tailored to cross-dataset ReID. They augment source domain data with perturbations generated from a domain adversarial network, improving the feature extractor's robustness to domain shifts. In this work, we build upon the idea of the domain adversarial network, aiming to improve the feature extractor's robustness to class space changes by the proposed class adversarial network.

**Generalizable Person Re-Identification.** DG methods for cross-dataset ReID can be categorized into two groups based on their use of source datasets (Choi et al., 2021). In the multi-source setting, multiple-source datasets are utilized to develop a generalizable model. Song et al. (2019) introduced a Domain-Invariant Mapping Network (DIMN) to establish a mapping between a person's image and its ID classifier. Jia et al. (2019) proposed the DualNorm approach, which combines Batch Normalization (Ioffe & Szegedy, 2015) and Instance Normalization (Ulyanov et al., 2016) to mitigate domain shift. The single-source setting is more challenging since only one source dataset is utilized. SNR (Jin et al., 2020) disentangles identity-relevant and identity-irrelevant features to reconstruct more generalizable features. DDAIG (Zhou et al., 2020b) augments source domain data with a domain adversarial network to increase training data quantity and diversity. MixStyle (Zhou et al., 2021b) mixes the feature statistics of instances to synthesize novel domains.

## 3 METHODOLOGY

### 3.1 PRELIMINARIES

**Heterogeneous Domain Generalization.** Let $\mathcal{X}$ denote an input feature space with dimension $d$ and $\mathcal{Y}$ a target class space. A domain is composed of data sampled from a distribution $\mathcal{D}$, where $\mathcal{D} = (x_i, y_i)_{i=1}^{n} \sim P_{(X,Y)}, x \in \mathcal{X} \subset \mathbb{R}^d, y \in \mathcal{Y} \subset \mathbb{R}$ and $n$ is the number of data in the domain. Here, $P_{(X,Y)}$ denotes the joint distribution of the input sample and class label, where $X$ and $Y$ denote the corresponding random variables. For the task of heterogeneous domain generalization, the input is $N$ source domains, $\mathcal{S} = \left\{ \mathcal{D}^i \mid i = 1, \cdots, N \right\}$, where $\mathcal{D}^i = \left\{ \left( x_j^i, y_j^i \right) \right\}_{j=1}^{n_i}$ denotes the $i_{th}$ domain and $\mathcal{Y}^i$ denotes the class space of the $i_{th}$ domain. Note that $P_{(X,Y)}^{(i)} \neq P_{(X,Y)}^{(i')}, \mathcal{Y}^i = \mathcal{Y}^{i'}, i \neq i'$ and $i, i' \in \{1, \cdots, N\}$. The goal of HDG is to learn a generalizable feature extractor $f(\cdot)$ from the $N$ source domains to achieve a minimum recognition error on an unseen target domain $\mathcal{T}$, where $\mathcal{T}$ cannot be accessed during training and $P_{(X,Y)}^{(\mathcal{T})} \neq P_{(X,Y)}^{(i)}, \mathcal{Y}^{\mathcal{T}} \neq \mathcal{Y}^i$ for $i \in \{1, \cdots, N\}$.

**HDG on Cross-Dataset ReID.** A dataset can encompass multiple domains in both single-source or multi-source settings. Within the ReID context, domains correspond to the data sampled from different cameras.

**Domain Adversarial Network (DAN).** The adversarial generation of domain perturbations has shown efficacy in addressing conventional DG challenges (Shankar et al., 2018; Zhou et al., 2020b). These methods enhance the feature extractor's robustness to domain shifts by augmenting source domain data with perturbations generated by domain generators. Figure 2 illustrates the structure of DAN, where the goal of the domain generator is to transform the input data in a manner that remains recognizable to the feature extractor while being misleading to the domain discriminator. In contrast, the domain discriminator aims to accurately classify the domains to which the original data and its domain-augmented version belong.

Let $T_\theta$ denote the domain generator, Equation 1 illustrates the generation of domain augmented data, denoted as $\hat{x}$, by combining the original input data $x$ with the domain perturbation generated by $T_\theta$. The perturbation weight, $\lambda$, controls the influence of domain perturbations.

$$\hat{x} = x + \lambda T_\theta(x) \tag{1}$$

The objective function for $T_\theta$ is expressed in Equation 2. Here, $f_\phi$ denotes the feature extractor, $f_\varphi$ denotes the domain discriminator, $\hat{\ell}_D$ denotes the cross-entropy loss of $f_\varphi$ for domain classification, and $\hat{\ell}_F$ denotes the cross-entropy loss of $f_\phi$ for category classification.

$$\min_\theta \hat{\ell}_F(f_\phi(\hat{x}), y) - \hat{\ell}_D(f_\varphi(\hat{x}), d) \tag{2}$$

The domain discriminator is specifically designed to capture domain-discriminative features. Its learning objective is to minimize the domain classification loss for $x$ and $\hat{x}$ with respect to $\varphi$,

$$\min_{\varphi} \hat{\ell}_D(f_\varphi(\hat{x}), d) + \ell_D(f_\varphi(x), d). \tag{3}$$

In DG, the architecture of the domain adversarial network differs slightly from the traditional GAN framework. Firstly, while the traditional GAN discriminator performs binary classification to verify whether the generated data is real or synthetic, the domain discriminator performs multi-class classification to identify the data domain, forcing the generator to synthesize data across multiple domain distributions instead of mapping the data to a singular domain distribution (Zhou et al., 2020b). Secondly, whereas the traditional GAN generator solely maximizes the classification loss determined by the discriminator, the domain generator additionally minimizes the classification loss determined by the feature extractor to ensure the synthetic data is semantically meaningful (Zhou et al., 2020b).

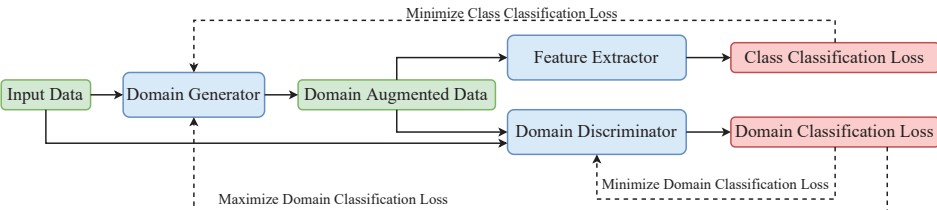

Figure 2: The domain generator maximizes the domain classification loss determined by the domain discriminator while minimizing the class classification loss determined by the feature extractor. Conversely, the domain discriminator minimizes the domain classification loss for both the original input data and its domain-augmented version.

### 3.2 ADAPTIVE ADVERSARIAL AUGMENTATION

To address the inconsistent class space challenges in applying DG methods to ReID tasks, we propose an HDG approach, Adaptive Adversarial Augmentation (AAA), for single-source cross-dataset ReID. Figure 3 illustrates the core concepts of AAA, contrasting it with the existing DAN-based approach. Unlike existing methods that solely perturb domain spaces, AAA concurrently trains the feature extractor alongside a DAN and a Class Adversarial Network (CAN). Beyond domain perturbation, CAN adversarially generates class perturbations to augment source domain data, improving the feature extractor's robustness to class space changes. Additionally, diverging from existing methods that apply a fixed perturbation impact factor, AAA adopts a diversity-based perturbation impact factor that dynamically adjusts the perturbation's effect based on the diversity of learned embeddings, thus providing a flexible augmentation strategy. Lastly, while existing methods only adopt cross-entropy loss to optimize the feature extractor (Zhou et al., 2020b), AAA also encompasses triplet loss (Schroff et al., 2015), which optimizes the embedding space such that data points with the same class are closer to each other than those with different classes (Hermans et al., 2017).

**Feature Extractor.** The feature extractor $f_\phi$ is designed to capture domain and class invariant representations by optimizing the cross-entropy and triplet loss for the input data $x$ and its domain-class augmented variant, $\tilde{x}$. Equation 4 delineates the generation of $\tilde{x}$. Here, $T_\theta$ and $T_\vartheta$ represent the domain and class generators, and $\lambda$ indicates the perturbation impact factor.

$$\tilde{x} = x + \lambda T_\theta(x) + \lambda T_\vartheta(x) \tag{4}$$

Let $x$ and $\tilde{x}$ denote the anchor data, with examples sharing the same class label as $x$ and $\tilde{x}$ serving as positive samples, denoted as $x_p$ and $\tilde{x}_p$, and examples possessing different class labels serving as negative samples, denoted as $x_n$ and $\tilde{x}_n$. Beyond minimizing cross-entropy loss, the feature extractor also minimizes the distance between the anchor data and positive samples while maximizing the distance between the anchor data and negative samples. Equation 5 presents the computation of the triplet loss, where $\tau$ is a margin hyperparameter that prevents trivial solutions by enforcing a minimum separation between positive and negative pairs.

$$\mathcal{L}_{\text{triplet}}(x, x_p, x_n) = \max\left(0, \|f(x) - f(x_p)\|^2 - \|f(x) - f(x_n)\|^2 + \tau\right) \tag{5}$$

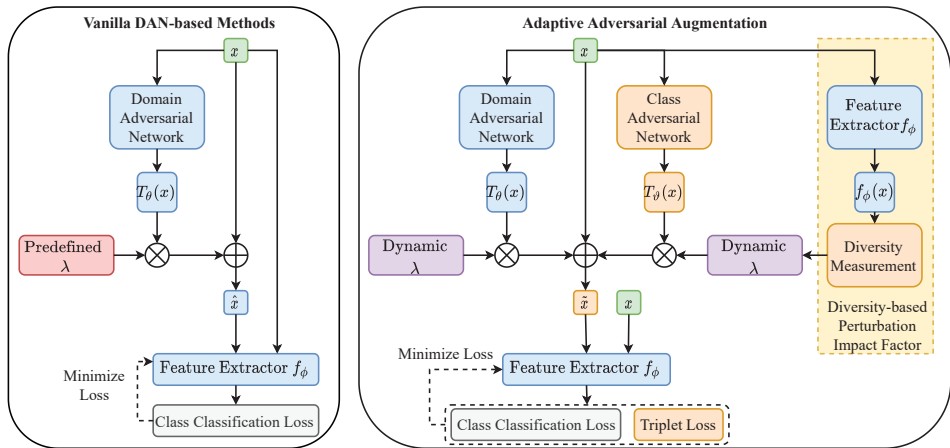

Figure 3: Comparison between Vanilla DAN-based methods and AAA. Here, $T_\theta(x)$ denotes the domain perturbation generated by DAN, $T_\vartheta(x)$ denotes the class perturbation generated by CAN, $\lambda$ denotes the perturbation impact factor, $\bar{x}$ denotes the domain-augmented data generated by the vanilla methods, and $\tilde{x}$ denotes the domain-class-augmented data generated by AAA.

Equation 6 presents the loss function of the feature extractor. Here, $\alpha$ represents a balance weight hyperparameter that controls the weight of the augmented data loss, whereas $\ell_F$ and $\tilde{\ell}_F$ represent the aggregate of cross-entropy and triplet loss for $x$ and $\tilde{x}$, respectively.

$$\min_{\phi} \quad \alpha\ell_F(f_\phi(x), y, f_\phi(x_p), f_\phi(x_n)) +$$
$$(1-\alpha)\tilde{\ell}_F(f_\phi(\tilde{x}), y, f_\phi(\tilde{x}_p), f_\phi(\tilde{x}_n)) \tag{6}$$

**Class Adversarial Network (CAN).** Equation 7 presents the generation of class augmented data, denoted as $\bar{x}$, by combining the original input $x$ with the class perturbation $T_\vartheta(x)$, where $\lambda$ controls the influence of class perturbations.

$$\bar{x} = x + \lambda T_\vartheta(x) \tag{7}$$

The objective function for $T_\vartheta$ is expressed in Equation 8. Here, $f_\psi$ denotes the class discriminator parameterized by $\psi$, $\bar{\ell}_C$ and $\bar{\ell}_F$ denote the cross-entropy loss of the class discriminator and feature extractor for class classification, respectively.

$$\min_{\vartheta} \bar{\ell}_F(f_\phi(\bar{x}), y) - \bar{\ell}_C(f_\psi(\bar{x}), y) \tag{8}$$

The class discriminator, $f_\psi$, is designed to capture class-discriminative features. Its learning objective is to minimize the class classification loss for $x$ and $\bar{x}$ with respect to $\psi$,

$$\min_{\psi} \bar{\ell}_C(f_\psi(\bar{x}), y) + \ell_C(f_\psi(x), y). \tag{9}$$

**Diversity-based Perturbation Impact Factor (DPIF).** As detailed in previous sections, our approach generates augmented data by combining original input $x$ with generated perturbations modulated by perturbation impact factors. Unlike existing methods that utilize a static perturbation impact factor, we introduce a diversity-based perturbation impact factor. DPIF quantifies the embedding diversity within each training batch and adjusts the perturbation impact factor accordingly, offering a more adaptive augmentation strategy.

Specifically, we observe that embeddings with high diversity indicate a sufficiently diverse dataset, where a large perturbation impact factor could be detrimental by over-perturbing model training and potentially hampering its generalization performance. Conversely, embeddings with low diversity suggest the need for a greater perturbation impact to enrich synthetic data diversity. DPIF measures the diversity of embeddings with the Gini coefficient, presented in Equation 10.

$$G = \frac{\sum_{i=1}^{n}\sum_{j=1}^{n}|f_\phi(x_i) - f_\phi(x_j)|}{2n^2\frac{1}{n}\sum_{i=1}^{n}f_\phi(x_i)} \tag{10}$$

Then, we calculate the base perturbation impact factor as $1 - G$. This is subsequently normalized into a range $k$, where $k$ is a diversity normalization factor hyperparameter constrained to $(0, 1]$. The computation of the diversity-based $\lambda$ is presented in Equation 11.

$$\lambda = k \times (1 - G) \tag{11}$$

In addition to the Gini coefficient, we explore an alternative method for assessing embedding diversity. Utilizing the K-means (Lloyd, 1982) clustering algorithm, we identify the centroid of the embeddings in the current training batch. We then compute distances between individual points and this centroid using metrics such as the Euclidean distance or Cosine similarity. The average of these distances serves as an indicator of embedding diversity. Further details are in Appendix A.4.

**Architecture Design.** Our framework utilizes ResNet-18 (He et al., 2016) as the backbone for the discriminators and the feature extractor. However, any network architecture suitable for the given problem can be utilized. To construct the generators, we leverage a Fully Convolutional Network (FCN) (Long et al., 2015) as it allows us to generate data efficiently, considering the high dimensionality of the data (Zhou et al., 2020b). The full algorithm is illustrated in Algorithm 1, and Appendix A.1 illustrates the training details of the feature extractor and class adversarial network.

---

**Algorithm 1** Adaptive Adversarial Augmentation

---

1: **Input:** $\mathcal{D}$: training set; $N_e$: maximum number of epochs.
2: **Output:** $f_\phi$: feature extractor.
3: **for** $i = 1$ to $N_e$ **do**
4:     Sample a batch of data, $(x, y, d) \sim \mathcal{D}$.
5:     Compute diversity-based perturbation impact factor with Eq. 11.
6:     Train domain adversarial network with Eq. 1, 2, 3.
7:     Train class adversarial network with Eq. 7, 8, 9.
8:     Generate domain-class augmented data with Eq. 4.
9:     Update feature extractor with Eq. 6.
10: **end for**

---

## 4 EXPERIMENTS

### 4.1 EXPERIMENT SETTING

**Datasets.** The proposed framework is evaluated on three widely used person ReID benchmark datasets. (1) The CUHK03 dataset (Li et al., 2014) comprises 14,096 images captured from five pairs of camera views. It includes 7,368 training images (767 identities), 1,400 query images (700 identities), and 5,328 gallery images. (2) The Market1501 dataset (Zheng et al., 2015) consists of 32,668 images captured from six cameras. It comprises 12,936 training images (751 identities), 3,368 query images (750 identities), and 19,732 gallery images. (3) The MSMT17 dataset (Wei et al., 2018) is a large-scale collection of images captured from 12 outdoor and three indoor cameras. It contains 126,441 training images (3,060 identities), 11,659 query images (3,060 identities), and 82,161 gallery images.

**Baselines.** We compare our framework with several state-of-the-art DG methods. These methods are CrossGrad (Shankar et al., 2018), DDAIG (Zhou et al., 2020b), MixStyle (Zhou et al., 2021b), DomainMix (Sun et al., 2022), and EFDMix (Zhang et al., 2022). We also evaluate classical data augmentation methods such as RandomErasing, RandomRotation, and ColorJitter. Furthermore, we include a baseline approach called Empirical Risk Minimization (ERM), which directly combines data from all source domains without employing DG techniques. Note that DDAIG and MixStyle are also designed for HDG.

**Evaluation Metrics.** We employ the single-source cross-dataset ReID evaluation strategy by following prior works (Zhou et al., 2020b;c; 2021b). Specifically, we designate one dataset as the source training set and evaluate the model using the query and gallery sets from the remaining datasets. We utilize two commonly used metrics in ReID tasks: mean Average Precision (mAP) and Top-$k$ accuracy. mAP evaluates the retrieval performance by calculating the average precision across all queries. Top-$k$ accuracy measures the model's ability to correctly identify the match within the top-$k$ retrieved results. In our evaluation, we report the top-1, top-5 and top-10 accuracy.

**Network Structure.** Images in each dataset are resized to $256 \times 128$ (Zhou et al., 2020b). We utilize the ResNet18 model pretrained on the ImageNet dataset as the backbone. During the testing phase, embeddings extracted from models are used to compute the Euclidean distance for image matching (Park & Ham, 2020). Section 7 presents the details of reproducibility.

## 4.2 EXPERIMENTAL RESULTS

This section presents the evaluation results on benchmark datasets, emphasising the best and second-best results, indicated by bolding and underlining, respectively. All experiments are run ten times, and the average results and standard deviation are reported.

**Evaluation on CUHK03.** Table 1 demonstrates the superior performance of AAA, surpassing both conventional data augmentation techniques and state-of-the-art DG methods. AAA consistently outperforms the second-best method across all target datasets, exhibiting substantial improvements in mAP, top-1, top-5, and top-10 accuracies. These evaluation results serve as compelling evidence of the effectiveness of synthetic data, which enhances the diversity of the source domain data.

Table 1: Evaluation on CUHK03.

| Method | Market1501 | | | | MSMT17 | | | |
|---|---|---|---|---|---|---|---|---|
| | mAP | top-1 | top-5 | top-10 | mAP | top-1 | top-5 | top-10 |
| ERM | 7.45 ± 0.44 | 19.93 ± 0.51 | 37.48 ± 0.69 | 46.42 ± 0.73 | 0.84 ± 0.10 | 3.15 ± 0.39 | 7.29 ± 0.77 | 10.06 ± 0.82 |
| Erasing | 8.49 ± 0.50 | 22.15 ± 0.51 | 40.14 ± 0.29 | 48.65 ± 0.21 | 0.88 ± 0.09 | 3.28 ± 0.28 | 7.56 ± 0.64 | 10.47 ± 0.80 |
| Rotation | 5.42 ± 0.38 | 15.93 ± 0.17 | 31.21 ± 0.32 | 39.40 ± 0.47 | 0.64 ± 0.10 | 2.39 ± 0.33 | 5.65 ± 0.67 | 7.90 ± 0.76 |
| ColorJitter | 2.15 ± 0.35 | 7.36 ± 0.38 | 16.64 ± 0.50 | 22.74 ± 0.66 | 0.20 ± 0.03 | 0.70 ± 0.19 | 1.92 ± 0.35 | 2.99 ± 0.46 |
| CrossGrad | 7.04 ± 0.43 | 19.34 ± 0.29 | 36.36 ± 0.27 | 44.99 ± 0.46 | 0.78 ± 0.11 | 2.98 ± 0.57 | 6.82 ± 0.62 | 9.48 ± 0.66 |
| DDAIG | 7.31 ± 0.44 | 19.89 ± 0.80 | 37.65 ± 0.96 | 46.35 ± 0.52 | 0.84 ± 0.15 | 3.33 ± 0.66 | 7.31 ± 0.55 | 10.16 ± 0.28 |
| MixStyle | 7.03 ± 0.73 | 19.06 ± 0.66 | 36.12 ± 0.27 | 44.81 ± 0.86 | 1.18 ± 0.04 | 4.91 ± 0.37 | 10.39 ± 0.48 | 13.98 ± 0.37 |
| DomainMix | 1.18 ± 0.13 | 4.17 ± 0.41 | 10.30 ± 0.39 | 15.01 ± 0.28 | 0.16 ± 0.01 | 0.58 ± 0.06 | 1.62 ± 0.19 | 2.46 ± 0.26 |
| EFDMix | 5.53 ± 0.71 | 16.16 ± 0.24 | 30.74 ± 0.26 | 38.50 ± 0.61 | 0.83 ± 0.14 | 3.43 ± 0.69 | 7.62 ± 0.97 | 10.26 ± 0.80 |
| AAA | **9.70 ± 0.31** | **25.83 ± 0.37** | **44.86 ± 0.31** | **53.33 ± 0.57** | **1.41 ± 0.11** | **5.70 ± 0.15** | **11.85 ± 0.22** | **15.77 ± 0.20** |

**Evaluation on Market1501.** The results presented in Table 2 reveal that both ERM and classical data augmentation methods surpass the performance of most DG methods. Figure 4 illustrates that the notable diversity inherent in the Market1501 dataset contributes to the superior performance achieved by classical methods. Among the DG methods evaluated, only AAA consistently achieves outstanding results across all target datasets. These findings in Table 2 indicate that existing DG methods may over-perturb the trained model, potentially compromising their generalization ability. Conversely, our diversity-based perturbation impact factor facilitates the modulation of perturbation magnitude in accordance with the diversity of learned embeddings, thereby enabling a more adaptable augmentation strategy and enhanced generalization performance.

Table 2: Evaluation on Market1501.

| Method | CUHK03 | | | | MSMT17 | | | |
|---|---|---|---|---|---|---|---|---|
| | mAP | top-1 | top-5 | top-10 | mAP | top-1 | top-5 | top-10 |
| ERM | 2.78 ± 0.50 | 2.21 ± 0.58 | 6.55 ± 0.68 | 10.01 ± 0.73 | 0.91 ± 0.10 | 3.15 ± 0.42 | 7.15 ± 0.74 | 9.89 ± 0.67 |
| Erasing | 3.33 ± 0.29 | 2.52 ± 0.35 | 7.24 ± 0.59 | 10.96 ± 0.71 | 1.07 ± 0.07 | 3.68 ± 0.34 | 8.03 ± 0.51 | 10.91 ± 0.72 |
| Rotation | 2.31 ± 0.72 | 1.51 ± 0.56 | 4.89 ± 0.78 | 7.66 ± 0.35 | 0.97 ± 0.06 | 3.39 ± 0.16 | 7.51 ± 0.62 | 10.20 ± 0.65 |
| ColorJitter | 0.39 ± 0.06 | 0.18 ± 0.06 | 0.81 ± 0.22 | 1.37 ± 0.31 | 0.23 ± 0.05 | 0.86 ± 0.24 | 2.26 ± 0.42 | 3.51 ± 0.64 |
| CrossGrad | 2.62 ± 0.24 | 1.95 ± 0.31 | 5.94 ± 0.42 | 9.06 ± 0.57 | 0.96 ± 0.14 | 3.39 ± 0.59 | 7.39 ± 0.90 | 10.21 ± 0.43 |
| DDAIG | 2.13 ± 0.32 | 1.69 ± 0.45 | 4.87 ± 0.59 | 7.76 ± 0.95 | 0.87 ± 0.12 | 3.19 ± 0.40 | 6.97 ± 0.82 | 9.45 ± 0.63 |
| MixStyle | 1.70 ± 0.37 | 1.30 ± 0.17 | 3.54 ± 0.26 | 5.87 ± 0.25 | 1.20 ± 0.31 | 4.56 ± 0.73 | 9.32 ± 0.75 | 12.45 ± 0.78 |
| DomainMix | 0.23 ± 0.03 | 0.05 ± 0.05 | 0.21 ± 0.15 | 0.57 ± 0.23 | 0.13 ± 0.01 | 0.43 ± 0.10 | 1.33 ± 0.22 | 2.13 ± 0.33 |
| EFDMix | 0.62 ± 0.31 | 0.27 ± 0.22 | 1.21 ± 0.78 | 2.07 ± 0.38 | 0.60 ± 0.25 | 2.45 ± 0.56 | 5.67 ± 0.96 | 7.94 ± 0.43 |
| AAA | **3.60 ± 0.46** | **3.00 ± 0.39** | **8.93 ± 0.12** | **13.57 ± 0.77** | **1.47 ± 0.24** | **5.10 ± 0.32** | **10.88 ± 0.61** | **14.25 ± 0.59** |

**Evaluation on MSMT17.** As shown in Table 3, AAA surpasses all baselines by a large margin across all target datasets. Notably, when the source dataset consists of a large number of domains and identities, AAA achieves a substantial performance improvement, nearly doubling the performance of the second-best model (CrossGrad). These results highlight the effectiveness of AAA in addressing the challenges posed by diverse and complex datasets.

Table 3: Evaluation on MSMT17.

| Method | CUHK03 | | | | Market1501 | | | |
|---|---|---|---|---|---|---|---|---|
| | mAP | top-1 | top-5 | top-10 | mAP | top-1 | top-5 | top-10 |
| ERM | 0.28 ± 0.07 | 0.12 ± 0.12 | 0.36 ± 0.12 | 0.69 ± 0.26 | 1.16 ± 0.30 | 3.55 ± 0.19 | 9.56 ± 0.26 | 14.21 ± 0.13 |
| Erasing | 0.26 ± 0.03 | 0.09 ± 0.07 | 0.33 ± 0.15 | 0.69 ± 0.17 | 1.10 ± 0.30 | 3.29 ± 0.58 | 8.91 ± 0.57 | 13.42 ± 0.23 |
| Rotation | 0.27 ± 0.03 | 0.09 ± 0.08 | 0.36 ± 0.11 | 0.93 ± 0.27 | 0.95 ± 0.08 | 2.77 ± 0.20 | 7.83 ± 0.62 | 11.89 ± 0.68 |
| ColorJitter | 0.24 ± 0.03 | 0.06 ± 0.03 | 0.38 ± 0.14 | 0.57 ± 0.18 | 0.66 ± 0.09 | 1.88 ± 0.31 | 5.64 ± 0.57 | 8.50 ± 0.42 |
| CrossGrad | 0.28 ± 0.05 | 0.07 ± 0.06 | 0.47 ± 0.28 | 0.87 ± 0.34 | 1.27 ± 0.31 | 3.83 ± 0.14 | 10.13 ± 0.61 | 14.89 ± 0.16 |
| DDAIG | 0.24 ± 0.02 | 0.01 ± 0.03 | 0.32 ± 0.12 | 0.67 ± 0.27 | 0.89 ± 0.12 | 2.81 ± 0.53 | 7.45 ± 0.87 | 11.14 ± 0.52 |
| MixStyle | 0.22 ± 0.03 | 0.02 ± 0.03 | 0.27 ± 0.21 | 0.50 ± 0.19 | 0.86 ± 0.21 | 2.56 ± 0.76 | 7.46 ± 0.75 | 11.37 ± 0.55 |
| DomainMix | 0.21 ± 0.03 | 0.01 ± 0.03 | 0.26 ± 0.17 | 0.48 ± 0.28 | 0.66 ± 0.09 | 1.91 ± 0.42 | 5.37 ± 0.78 | 8.32 ± 0.62 |
| EFDMix | 0.23 ± 0.03 | 0.05 ± 0.05 | 0.28 ± 0.14 | 0.47 ± 0.28 | 0.90 ± 0.22 | 2.74 ± 0.86 | 7.80 ± 0.89 | 11.66 ± 0.53 |
| AAA | **0.35 ± 0.03** | **0.29 ± 0.05** | **0.71 ± 0.17** | **1.21 ± 0.22** | **1.93 ± 0.20** | **6.89 ± 0.52** | **16.30 ± 0.73** | **22.12 ± 0.33** |

## 4.3 ABLATION STUDY

**Class Adversarial Network.** Table 4 displays results from training on the CUHK03 dataset for AAA, both with and without the Class Adversarial Network (CAN). The first row represents the results of DDAIG, as it is equivalent to AAA when the perturbation impact factor is fixed and the CAN module is removed. To ensure a fair comparison, we set $\lambda = 0.3$ by following (Zhou et al., 2020b) configuration. Please refer to Appendix A.2 for the results of models trained on Market1501 and MSMT17 datasets. Table 4 shows the effectiveness of the CAN module, which effectively improves the feature extractor's robustness to changes in the class space.

Table 4: Ablation Experiment for Class Adversarial Network on CUHK03.

| Method | Market1501 | | | | MSMT17 | | | |
|---|---|---|---|---|---|---|---|---|
| | mAP | top-1 | top-5 | top-10 | mAP | top-1 | top-5 | top-10 |
| DDAIG | 7.31 ± 0.44 | 19.89 ± 0.80 | 37.65 ± 0.96 | 44.99 ± 0.46 | 0.78 ± 0.11 | 2.98 ± 0.57 | 6.82 ± 0.62 | 9.48 ± 0.66 |
| AAA $\lambda = 0.3$ | **9.49 ± 0.31** | **24.73 ± 0.31** | **43.85 ± 0.53** | **52.76 ± 0.43** | **1.15 ± 0.05** | **4.87 ± 0.15** | **10.46 ± 0.39** | **13.90 ± 0.50** |

**Diversity-based Perturbation Impact Factor.** Table 5 presents the results obtained from training on the CUHK03 dataset by varying values of $\lambda$, within the range of 0.1 to 0.5. The normalization factor $k$ is set to 0.2. Please refer to Appendix A.3 for results pertaining to models trained on Market1501 and MSMT17 datasets. Table 5 illustrates that incorporating perturbations can improve the model's generalization ability. However, excessive perturbation can negatively impact the model's performance, while inadequate perturbation may not challenge the model sufficiently. Table 5 shows that the optimal value of $\lambda$ for CUHK03 lies between 0.1 and 0.3. Nonetheless, manual tuning of $\lambda$ is labour-intensive. By dynamically adjusting $\lambda$ based on the diversity of learned embeddings, AAA's performance surpasses that achieved with a fixed $\lambda$ value.

Table 5: Ablation Experiment for $\lambda$ on CUHK03.

| Method | Market1501 | | | | MSMT17 | | | |
|---|---|---|---|---|---|---|---|---|
| | mAP | top-1 | top-5 | top-10 | mAP | top-1 | top-5 | top-10 |
| $\lambda = 0.1$ | 9.64 ± 0.47 | 24.64 ± 0.13 | 43.05 ± 0.47 | 52.20 ± 0.51 | 1.35 ± 0.15 | 5.51 ± 0.23 | 11.25 ± 0.39 | 15.01 ± 0.26 |
| $\lambda = 0.3$ | 9.49 ± 0.31 | 24.73 ± 0.31 | 43.85 ± 0.53 | 52.76 ± 0.43 | 1.15 ± 0.05 | 4.87 ± 0.15 | 10.46 ± 0.39 | 13.90 ± 0.50 |
| $\lambda = 0.5$ | 7.35 ± 0.46 | 21.32 ± 0.44 | 38.33 ± 0.49 | 47.73 ± 0.67 | 1.01 ± 0.05 | 4.50 ± 0.27 | 9.76 ± 0.45 | 13.11 ± 0.58 |
| Dynamic $\lambda$ | **9.70 ± 0.31** | **25.83 ± 0.37** | **44.86 ± 0.31** | **53.33 ± 0.57** | **1.41 ± 5.70** | **5.70 ± 0.15** | **11.85 ± 0.22** | **15.77 ± 0.20** |

## 4.4 FURTHER ANALYSIS

**T-SNE Visualization.** To gain insights into the effectiveness of AAA, we apply t-SNE (Van der Maaten & Hinton, 2008) to visualize the feature embeddings in the domain space, as depicted in Figure 4. It is evident that in datasets with small diversity, such as CUHK03 and MSMT17, the synthetic data distributions filled previously unfilled spaces, exploring unseen domain and class spaces. This results in the model learning more domain-class-agnostic representations, which explains the superior performance of AAA on these datasets. On the other hand, in datasets with large diversity, like Market1501, the synthetic data distributions overlap with existing data. Consequently,

the synthetic data may over-perturb the model training procedure, leading to a potential decrease in its generalization ability. This highlights the need for careful consideration of perturbation impact factors when applying adversarial data generation methods. By incorporating a diversity-based perturbation impact factor, AAA also achieves superior performance on the Market1501 dataset.

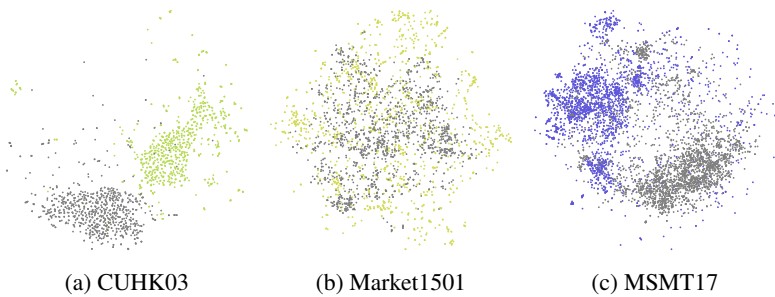

(a) CUHK03          (b) Market1501          (c) MSMT17

Figure 4: T-SNE Visualization for benchmark datasets. Coloured and grey points denote the original and augmented data, respectively.

**Perturbation Visualization.** Figure 5 illustrates the impact of different perturbations on image transformations. A comparison between the domain perturbation (2nd column) and class perturbation (4th column) reveals distinct effects. The domain perturbation, $T_\theta(x)$, primarily perturbs domain-related features such as the background. Conversely, class perturbation, $T_\vartheta(x)$, mainly perturbs class-related features, such as clothing and the human body. By incorporating both domain and class perturbations, the augmented images, $\tilde{x}$, exhibit intricate transformations for domain and class-related features, facilitating the model in capturing domain invariant representations and enhancing its robustness to class space changes. Please refer to Appendix A.5 for the visualization of Market1501 and MSMT17 datasets.

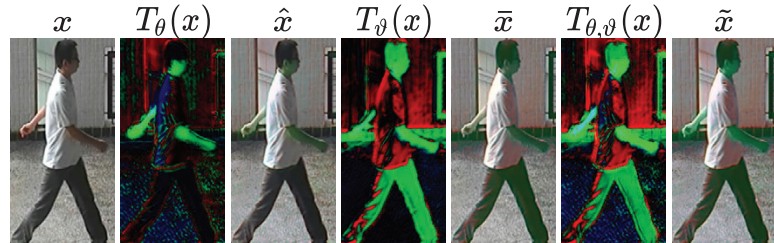

Figure 5: Examples of transformed images from CUHK03. Here, $x$ denotes the original image. $T_\theta(x)$ and $\hat{x}$ denote domain perturbation and domain augmented image. $T_\vartheta(x)$ and $\bar{x}$ denote class perturbation and class augmented image. $T_{\theta,\vartheta}(x)$ and $\tilde{x}$ denote combined perturbation and domain-class augmented image.

## 5 CONCLUSION

We proposed an HDG approach, Adaptive Adversarial Augmentation, to address the single-source cross-dataset person ReID problem. Unlike existing methods solely utilizing the domain adversarial networks, AAA incorporates a class adversarial network to enhance the feature extractor's adaptability to class space variations and its ability to identify novel classes in unseen domains. Moreover, we propose a diversity-based perturbation impact factor, yielding an adaptable augmentation strategy. Experiments across three popular person ReID benchmarks demonstrated AAA's superior effectiveness, surpassing state-of-the-art methods. Ablation studies further validate the effectiveness of each AAA component, and the visualization results deepened our understanding of AAA's applicability to the single-source cross-dataset ReID problem.

## 6 ETHIC STATEMENTS

This research endeavours to address the Heterogeneous Domain Generalization (HDG) challenge in the context of person re-identification (ReID), a field with many applications ranging from security surveillance to wildlife monitoring. The data employed in our experiments encompass three publicly available datasets, CUHK03, Market1501, and MSMT17, collected under ethical guidelines and with appropriate permissions where required. Our work adheres to the ethical guidelines stipulated by our institutions and the broader research community. We remain open to collaboration and constructive feedback to ensure the responsible advancement of knowledge in this domain.

## 7 REPRODUCIBILITY STATEMENT

**Code.** Our code is available at https://anonymous.4open.science/r/HDG-776E/. The code includes data preprocessing scripts, model implementation, and evaluation scripts. The code is well-commented, organized in a modular fashion, and accompanied by a README file explaining how to execute the code to reproduce the paper's results.

**Data.** Our experiments utilize the CUHK03, Market1501, and MSMT17 datasets, which are publicly available. We provide scripts to download and preprocess these datasets to the required format for our experiments.

**Evaluation.** The evaluation metrics and procedures are clearly defined in the paper. Our code will include scripts to evaluate the models.

**Dependencies.** Our framework is implemented using the Pytorch and Dassl (Zhou et al., 2020b) libraries.

**Computational Resources.** All experiments are conducted on NVIDIA Tesla A100 GPUs.

**Hyperparameters.** We use Stochastic Gradient Descent (SGD) as the optimizer with a momentum of 0.9 and weight decay of 5e-4. The normalization range $k$ for all datasets is set to 0.2. The balance weight $\alpha$ is set to 0.5. The margin $\tau$ for triplet loss is set to 1.2. All models are trained for 200 epochs with five epochs warmup. For CUHK03 and MSMT17 datasets, the batch size is set to 256 (64 identities and four images for each identity). For Market1501, the batch size is set to 64 (16 identities and four images for each identity). The learning rate for CUHK03 and MSMT17 is set to 1e-4, and for Market1501, it is set to 2.5e-4.

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

# A    APPENDIX

We present a detailed illustration of the training procedure for feature extractor and class adversarial network in Appendix A.1, additional results of ablation studies for class adversarial network in Appendix A.2, additional results of ablation studies for diversity-based perturbation impact factor in Appendix A.3, evaluation results for different types of diversity measures in Appendix A.4, and the perturbation visualization for Market1501 and MSMT17 datasets in Appendix A.5.

## A.1    ILLUSTRATION OF TRAINING PROCEDURE

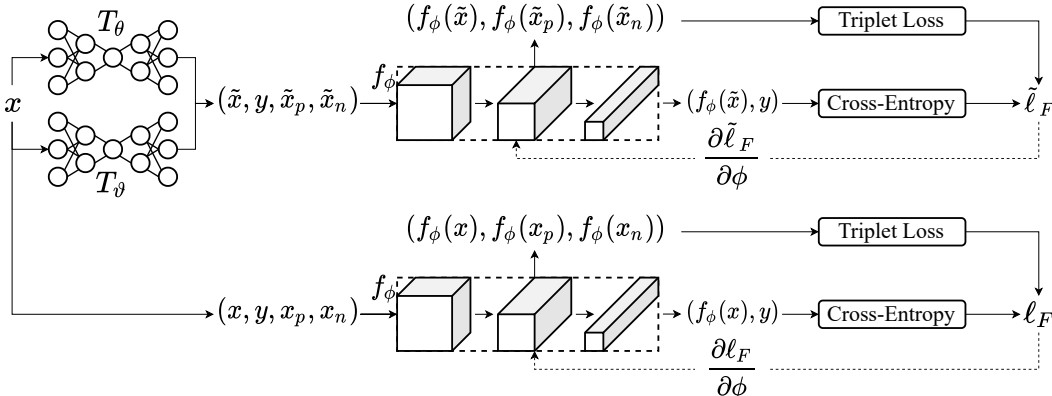

Figure 6: Training Feature Extractor $f_\phi$. AAA first generates $\tilde{x}$ by augmenting the original input $x$ with domain and class perturbations. Subsequently, the feature extractor minimizes the class classification cross-entropy loss and triplet loss for both $x$ and $\tilde{x}$.

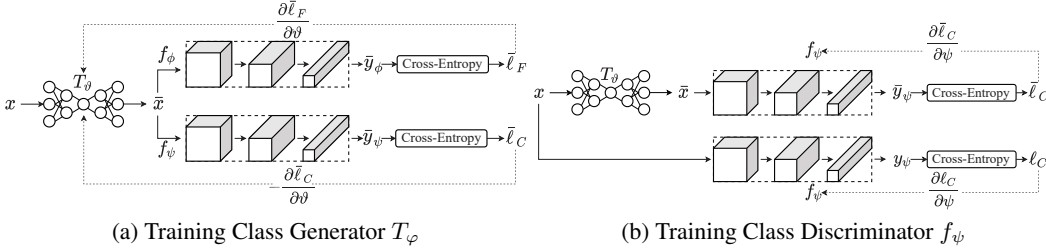

(a) Training Class Generator $T_\varphi$                    (b) Training Class Discriminator $f_\psi$

Figure 7: The class generator maximizes the class classification loss, $\bar{\ell}_C$, as determined by the class discriminator, while minimizing the class classification loss, $\bar{\ell}_F$, determined by the feature extractor to ensure generating semantically meaningful data. Conversely, the class discriminator minimizes the class classification loss for both $x$ and $\bar{x}$.

## A.2    CLASS ADVERSARIAL NETWORK

Tables 6 and 7 present results from training on the Market1501 and MSMT17 datasets with and without the Class Adversarial Network (CAN). These findings are consistent with the results discussed in the paper, further validating the effectiveness of CAN.

Table 6: Ablation Experiment for Class Adversarial Network on Market1501.

| Method | CUHK03 | | | | MSMT17 | | | |
|---|---|---|---|---|---|---|---|---|
| | mAP | top-1 | top-5 | top-10 | mAP | top-1 | top-5 | top-10 |
| DDAIG | $2.13 \pm 0.32$ | $1.69 \pm 0.45$ | $4.87 \pm 0.59$ | $7.76 \pm 0.95$ | $0.87 \pm 0.12$ | $3.19 \pm 0.40$ | $6.97 \pm 0.82$ | $9.45 \pm 0.63$ |
| AAA $\lambda = 0.3$ | $\mathbf{2.92 \pm 0.33}$ | $\mathbf{2.36 \pm 0.39}$ | $\mathbf{7.21 \pm 0.62}$ | $\mathbf{10.93 \pm 0.77}$ | $\mathbf{1.30 \pm 0.13}$ | $\mathbf{4.37 \pm 0.16}$ | $\mathbf{8.91 \pm 0.37}$ | $\mathbf{13.78 \pm 0.52}$ |

Table 7: Ablation Experiment for Class Adversarial Network on MSMT17.

| Method | CUHK03 | | | | Market1501 | | | |
|---|---|---|---|---|---|---|---|---|
| | mAP | top-1 | top-5 | top-10 | mAP | top-1 | top-5 | top-10 |
| DDAIG | 0.24 ± 0.02 | 0.01 ± 0.03 | 0.32 ± 0.12 | 0.67 ± 0.27 | 0.89 ± 0.21 | 2.81 ± 0.53 | 7.45 ± 0.87 | 11.14 ± 0.52 |
| AAA $\lambda = 0.3$ | **0.30 ± 0.04** | **0.14 ± 0.05** | **0.61 ± 0.19** | **1.00 ± 0.28** | **1.44 ± 0.19** | **5.26 ± 0.33** | **12.92 ± 0.45** | **19.83 ± 0.83** |

## A.3 DIVERSITY-BASED PERTURBATION IMPACT FACTOR

Tables 8 and 9 corroborate the efficacy of our proposed diversity-based perturbation impact factor, aligning with the findings discussed in the paper.

Table 8: Ablation Experiment for $\lambda$ on Market1501.

| Method | CUHK03 | | | | MSMT17 | | | |
|---|---|---|---|---|---|---|---|---|
| | mAP | top-1 | top-5 | top-10 | mAP | top-1 | top-5 | top-10 |
| $\lambda = 0.1$ | 3.32 ± 0.26 | 2.79 ± 0.31 | 8.07 ± 0.48 | 11.79 ± 0.51 | 1.27 ± 0.16 | 5.02 ± 0.23 | 9.89 ± 0.56 | 12.88 ± 0.67 |
| $\lambda = 0.3$ | 2.92 ± 0.33 | 2.36 ± 0.39 | 7.21 ± 0.62 | 10.93 ± 0.77 | 1.46 ± 0.21 | 5.08 ± 0.28 | 10.53 ± 0.44 | 14.08 ± 0.46 |
| $\lambda = 0.5$ | 2.70 ± 0.42 | 2.21 ± 0.36 | 6.86 ± 0.20 | 8.89 ± 0.47 | 1.30 ± 0.13 | 4.37 ± 0.16 | 8.91 ± 0.37 | 13.78 ± 0.52 |
| Dynamic $\lambda$ | **3.60 ± 0.46** | **3.00 ± 0.39** | **8.93 ± 0.12** | **13.57 ± 0.77** | **1.47 ± 0.24** | **5.10 ± 0.32** | **10.88 ± 0.61** | **14.25 ± 0.59** |

Table 9: Ablation Experiment for $\lambda$ on MSMT17.

| Method | CUHK03 | | | | Market1501 | | | |
|---|---|---|---|---|---|---|---|---|
| | mAP | top-1 | top-5 | top-10 | mAP | top-1 | top-5 | top-10 |
| $\lambda = 0.1$ | 0.33 ± 0.03 | 0.27 ± 0.10 | 0.64 ± 0.11 | 1.07 ± 0.15 | 1.63 ± 0.20 | 5.49 ± 0.29 | 13.27 ± 0.73 | 21.00 ± 0.56 |
| $\lambda = 0.3$ | 0.30 ± 0.04 | 0.14 ± 0.05 | 0.61 ± 0.19 | 1.00 ± 0.28 | 1.44 ± 0.19 | 5.26 ± 0.33 | 12.92 ± 0.45 | 19.83 ± 0.83 |
| $\lambda = 0.5$ | 0.27 ± 0.03 | 0.12 ± 0.07 | 0.43 ± 0.15 | 0.86 ± 0.23 | 1.38 ± 0.17 | 4.90 ± 0.47 | 12.08 ± 0.41 | 16.60 ± 0.80 |
| Dynamic $\lambda$ | **0.35 ± 0.03** | **0.29 ± 0.05** | **0.71 ± 0.17** | **1.21 ± 0.22** | **1.93 ± 0.20** | **6.89 ± 0.52** | **16.30 ± 0.73** | **22.12 ± 0.33** |

## A.4 OTHER DIVERSITY MEASURES

Tables 10, 11, 12 demonstrate the evaluation results of AAA with different diversity Measures. These results further validate the effectiveness of the proposed adaptive approach.

Table 10: Different Diversity Measures on CUHK03

| Method | Market1501 | | | | MSMT17 | | | |
|---|---|---|---|---|---|---|---|---|
| | mAP | top-1 | top-5 | top-10 | mAP | top-1 | top-5 | top-10 |
| Euclidean | 9.66 ± 0.39 | 25.30 ± 0.73 | 43.97 ± 0.74 | 52.29 ± 0.57 | 1.35 ± 0.15 | 5.44 ± 0.51 | 11.63 ± 0.42 | 15.52 ± 0.36 |
| Cosine | 9.58 ± 0.46 | 24.91 ± 0.71 | 43.74 ± 0.64 | 52.08 ± 0.85 | 1.28 ± 0.14 | 5.19 ± 0.47 | 11.27 ± 0.51 | 14.33 ± 0.29 |
| Gini | 9.70 ± 0.31 | 25.83 ± 0.37 | 44.86 ± 0.31 | 53.33 ± 0.57 | 1.41 ± 0.11 | 5.70 ± 0.15 | 11.85 ± 0.22 | 15.77 ± 0.20 |

Table 11: Different Diversity Measures on Market1501

| Method | CUHK03 | | | | MSMT17 | | | |
|---|---|---|---|---|---|---|---|---|
| | mAP | top-1 | top-5 | top-10 | mAP | top-1 | top-5 | top-10 |
| Euclidean | 3.21 ± 0.23 | 2.64 ± 0.26 | 7.57 ± 0.42 | 11.64 ± 0.84 | 1.41 ± 0.22 | 4.96 ± 0.46 | 9.56 ± 0.70 | 13.17 ± 0.41 |
| Cosine | 3.18 ± 0.38 | 2.36 ± 0.22 | 7.50 ± 0.39 | 11.57 ± 0.79 | 1.35 ± 0.16 | 4.85 ± 0.31 | 9.69 ± 0.39 | 13.22 ± 0.44 |
| Gini | 3.60 ± 0.46 | 3.00 ± 0.39 | 8.93 ± 0.12 | 13.57 ± 0.77 | 1.47 ± 0.24 | 5.10 ± 0.32 | 10.88 ± 0.61 | 14.25 ± 0.59 |

Table 12: Different Diversity Measures on MSMT17

| Method | CUHK03 | | | | Market1501 | | | |
|---|---|---|---|---|---|---|---|---|
| | mAP | top-1 | top-5 | top-10 | mAP | top-1 | top-5 | top-10 |
| Euclidean | 0.33 ± 0.03 | 0.14 ± 0.08 | 0.64 ± 0.12 | 1.07 ± 0.17 | 1.77 ± 0.20 | 5.94 ± 0.77 | 13.42 ± 0.56 | 19.09 ± 0.59 |
| Cosine | 0.31 ± 0.03 | 0.14 ± 0.05 | 0.62 ± 0.13 | 1.17 ± 0.13 | 1.76 ± 0.24 | 6.06 ± 0.44 | 13.60 ± 0.69 | 18.26 ± 0.38 |
| Gini | 0.35 ± 0.03 | 0.29 ± 0.05 | 0.71 ± 0.17 | 1.21 ± 0.22 | 1.93 ± 0.20 | 6.89 ± 0.52 | 16.30 ± 0.73 | 22.12 ± 0.33 |

## A.5  PERTURBATION VISUALIZATION

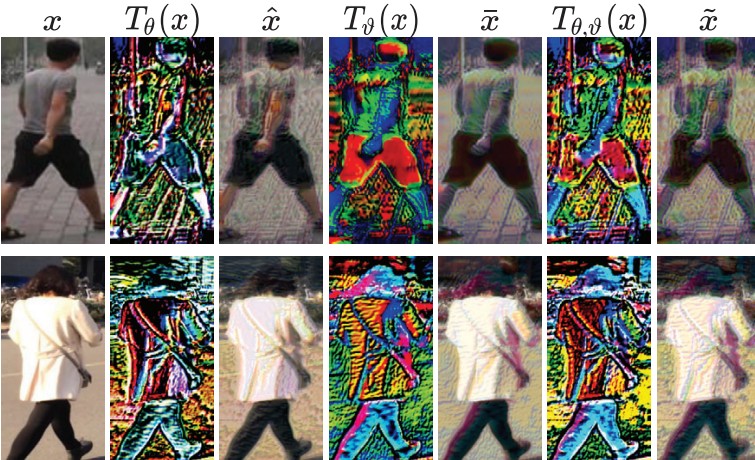

$$x \qquad T_\theta(x) \qquad \hat{x} \qquad T_\vartheta(x) \qquad \bar{x} \qquad T_{\theta,\vartheta}(x) \qquad \tilde{x}$$

Figure 8: Examples of transformed images from Market1501 (1st row) and MSMT17 (2nd row). Here, $x$ denotes the original image. $T_\theta(x)$ and $\hat{x}$ denote domain perturbation and domain augmented image. $T_\vartheta(x)$ and $\bar{x}$ denote class perturbation and class augmented image. $T_{\theta,\vartheta}(x)$ and $\tilde{x}$ denote combined perturbation and final augmented image.

