# OpenReview forum: "Heterogeneous Domain Generalization for Single-Source Cross-Dataset Person ReID: An Adaptive Adversarial Augmentation Approach"
_ICLR.cc/2024/Conference — ICLR 2024 Conference Withdrawn Submission_

### Official Review · Reviewer_cRe1 · 2023-10-18

**Soundness:** 3 good
**Presentation:** 2 fair
**Contribution:** 2 fair
**Rating:** 5
**Confidence:** 4

**Summary:**

The paper introduces Adaptive Adversarial Augmentation (AAA), a novel approach for addressing the issue of performance degradation in supervised person re-identification (ReID) models when applied to unseen domains. While existing domain generalization (DG) methods assume consistent class spaces between source and target domains, AAA is a Heterogeneous Domain Generalization (HDG) method tailored for single-source cross-dataset ReID.

AAA jointly trains a feature extractor, a Domain Adversarial Network (DAN), and a Class Adversarial Network (CAN) to enhance the feature extractor's robustness to both domain shifts and changes in class spaces. Additionally, the paper introduces a diversity-based perturbation impact factor to dynamically adjust the augmentation strategy based on the diversity of learned embeddings.

Experimental results indicate that AAA outperforms state-of-the-art methods on large-scale cross-dataset ReID benchmarks. In summary, the paper's contributions are the development of an HDG approach, the introduction of DAN and CAN, and the dynamic augmentation strategy, all of which lead to improved ReID performance in heterogeneous domains.

**Strengths:**

1. The authors appear to address a more challenging problem: single-source domain cross-dataset ReID. They introduce Adaptive Adversarial Augmentation (AAA), a Heterogeneous Domain Generalization approach, all without access to data from the target domain.
2. On three benchmark datasets, AAA outperforms the current state-of-the-art Domain Generalization (DG) and Heterogeneous Domain Generalization (HDG) methods.
3. The author provides open-source code that may be useful to the ReID community.

**Weaknesses:**

1. The paper's presentation left room for improvement, particularly in terms of clarifying its contributions. Figure 1 broadly illustrates the motivation behind the proposed method. However, the distinction between cross-dataset ReID and supervised ReID is widely known. To strengthen the paper's motivation, it is advisable to delineate the variances between single-source cross-data, single-source UDA, single-source DG, and multi-source DG, as presented in Figure 1 of the referenced literature [1]. Furthermore, the authors' contributions require clearer enumeration. The section on UDA methods should be concise, with a greater focus on the paper's unique contributions.

2. The level of innovation in this work appears to be somewhat incremental. The authors' CAN within the AAA framework bears a striking resemblance to DAN. Equations 7 and 8 are essentially identical to Equations 1 and 2, with the sole replacement of "domain" with "class."

3. The experimental aspect of the paper exhibits some shortcomings. Firstly, it fails to compare the proposed method with the latest DG techniques, such as GMoE [2]. Secondly, relying solely on ResNet is insufficient. It is crucial to demonstrate the applicability of the method to ViT.



Suggestions:
1. For the author's method name abbreviation, it may be that A^3 is better compared to AAA.

**Reference**

[1] Yuyang Zhao, Zhun Zhong, Fengxiang Yang, Zhiming Luo, Yaojin Lin, Shaozi Li, and Nicu Sebe. Learning to generalize unseen domains via memory-based multi-source meta-learning for person re-identification. In Proceedings of the IEEE/CVF Conference on Computer Vision and Pattern Recognition, pp. 6277–6286, 2021

[2] Li, Bo, Yifei Shen, Jingkang Yang, Yezhen Wang, Jiawei Ren, Tong Che, Jun Zhang, and Ziwei Liu. "Sparse Mixture-of-Experts are Domain Generalizable Learners." In The Eleventh International Conference on Learning Representations. 2023.

**Questions:**

1. **Evaluating Unsupervised Methods**: Given the significant challenges faced in single-source cross-dataset ReID and the relatively low performance of existing methods (For instance, the performance does not exceed 1% on MSMT17. ), it is worth exploring the potential of unsupervised methods. Unsupervised learning, where models learn from unlabeled data, might provide better adaptability to unseen datasets. A question to consider is whether unsupervised ReID methods, which rely on domain adaptation, can surpass the limitations of supervised methods and achieve higher accuracy in cross-dataset scenarios. Investigating the efficacy of unsupervised learning approaches in this context could shed light on new directions for ReID research.

2. **Generalized vs. Tailored Approaches**: The authors' proposed method, as observed, seems to have a broader scope as a generalized cross-dataset approach, rather than being tailored specifically for ReID. While this generality is valuable for cross-domain scenarios, it's essential to assess its performance in comparison to ReID-specific techniques. Therefore, an interesting question emerges: How does the authors' method perform in a head-to-head comparison with state-of-the-art ReID algorithms designed explicitly for the task? Evaluating its performance in the context of domain adaptation, along with conventional ReID models, would help clarify the method's strengths and limitations in ReID scenarios.

3. **Evaluation with DomainBed**: DomainBed is a versatile framework for domain generalization and domain adaptation tasks. Given the authors' method's broader applicability, it is pertinent to inquire about its performance when rigorously tested within the DomainBed framework. Assessing how the authors' method handles domain adaptation and generalization challenges, along with benchmarking against other specialized methods within DomainBed, could provide valuable insights into its adaptability and comparative effectiveness across diverse domains.

These questions delve deeper into the challenges, scope, and implications of the authors' work and can serve as a basis for further discussion and exploration in the field of cross-dataset ReID and domain adaptation in computer vision.

---

> ### Author Response · Authors · 2023-11-22
>
> Weakness - Improve Presentation and Clarify Contribution: In the revised version, we will follow the suggestions to improve the presentation of Figure 1. We will also concise the UDA section.
>
> Weakness - Insufficient Experiment: We will also evaluate our method in the context of classic Domain Generalization and compare it with the state-of-the-art DG methods.

---

> > ### Comment · Reviewer_cRe1 · 2023-11-22
> > **Didn't see the revised paper**
> >
> > Thanks to the author for the reply. It's great to see that the authors were able to revise the paper based on the suggestions, but I clicked on REVISION and didn't see the revised version.ICLR allows you to submit the revised version directly and generally highlights the changes in red.

---

> > > ### Author Response · Authors · 2023-11-23
> > >
> > > Sorry for the mislead. We will add all the stuff in the revised version and resubmit it next time.
> > > Thanks for your valuable comments.

---

### Official Review · Reviewer_nbhP · 2023-10-25

**Soundness:** 3 good
**Presentation:** 3 good
**Contribution:** 2 fair
**Rating:** 3
**Confidence:** 5

**Summary:**

This paper proposed an HDG approach, Adaptive Adversarial Augmentation (AAA) to address the single-source cross-dataset person ReID problem. Based on the domain adversarial networks, AAA incorporates a class adversarial network.

**Strengths:**

The addition of the class adversarial network seems to have achieved some effect.
The writing of the article is smooth and easy to understand.

**Weaknesses:**

The novelty is limited.The addition of CAN with a similar structure to DAN seems to be an incremental effort. In addition, the combination of cross-entropy loss and triplet loss is very common in the field of traditional deep learn-based person re-identification.

The related work only introduced the methods of generalizable person re-identification before 2021, and lacks the introduction of the latest methods.

The experimental part only compares  some baseline methods in generalization fields, and lacks the comparison of the generalizable person re-identification methods.

**Questions:**

See the weaknesses.

---

> ### Author Response · Authors · 2023-11-22
>
> Weakness - Related Work Outdated: In the revised version, we will add the update-to-date literature.
>
> Weakness - Insufficient Experiment Comparison: In the revised version, we will compare the proposed method with the state-of-the-art generalizable ReID baseline.

---

### Official Review · Reviewer_yLHp · 2023-10-30

**Soundness:** 3 good
**Presentation:** 2 fair
**Contribution:** 2 fair
**Rating:** 3
**Confidence:** 4

**Summary:**

Briefly summarize the paper and its contributions. This is not the place to critique the paper; the authors should generally agree with a well-written summary. Based on the available DAN, they further propose CAN and a dynamic diversity-based perturbation impact factor, together called AAA. Their experiments on multiple datasets proved the effectiveness of their methods.

**Strengths:**

(+) The proposed method is clear and easy to understand.

**Weaknesses:**

(-) [Incremental novelty] Comparing Eqs. (1) and (4), the innovations seem to be a limited improvement over the base method (DAN). The proposed CAN+dynamic lambda is more like a refined regularization constraint on the DAN, and does not jump out of the DAN's scope. More explanation from the author is needed to highlight their differences and contributions.
(-) [Incremental novelty] Following the previous comment, although the authors analyze in the experimental section (Figure 5) that DAN and CAN focus on different background-related and pedestrian-related regions, respectively, it is still possible to see that there are also pedestrian-highlighted regions in DAN's map (similar to CAN). It makes me more likely to stick to my previous comment.
(-) [Unclear definition] The definition in section 3.1 is not detailed and reader-friendly enough. (a) The explanation of  \mathcal{Y}^i appears before the usage of it. (b) What is the role of the joint probability distribution P_(X,Y)? It doesn't seem to appear after the definition.
(-) [Experiment] In the abstract, the authors emphasize that AAA is specifically designed for the single-source setting. At the method level, however, the proposed method has little relevance to single- or multi-domain settings, because eventually, the authors modeled domain features with a camera. Therefore I am curious about the performance of the proposed AAA in a multi-source domain setting, or the reasons why experiments in the multi-source domain setting cannot be carried out.

**Questions:**

See weaknesses.

---

> ### Author Response · Authors · 2023-11-22
>
> Weakness - Unclear Definition: In the revised version, we will improve our presentation of definitions. We will move the explanations after the usage of notation and clarify the role of the joint probability distribution.
>
> Weakness - Experiment: The definition of multiple domains(camera) and single source(dataset) may lead people confused about them. In the revised version, we will explicitly clarify the difference between. Also, we will evaluate the effectiveness of our method in the multi-source dataset setting.

---

### Official Review · Reviewer_hNEm · 2023-10-31

**Soundness:** 3 good
**Presentation:** 2 fair
**Contribution:** 2 fair
**Rating:** 3
**Confidence:** 4

**Summary:**

The paper introduces Adaptive Adversarial Augmentation (AAA) for Heterogeneous Domain Generalization in single-source cross-dataset person re-identification. AAA jointly trains a feature extractor, Domain Adversarial Network (DAN), and Class Adversarial Network (CAN) to enhance robustness to domain shifts and class space changes. Experimental results show that AAA outperforms state-of-the-art methods in large-scale cross-dataset person re-identification benchmarks.

**Strengths:**

1. The authors try to solve the most challenging setting in ReID tasks-Single-source cross-dataset ReID that needs to train a model on one source dataset to generalize across multiple unseen target datasets.

2. To solve the above challenge task, the authors propose Adaptive Adversarial Augmentation (AAA), a Heterogeneous Domain Generalization method.

**Weaknesses:**

1. Validity of the proposed methodology. Since the compared methods all perform poorly, AAA does outperform the others in comparison, but AAA's performance is equally poor, which casts doubt on whether AAA is really effective.

2. Innovatively it seems to be incremental.AAA compared to the Vanilla-DAN-based method looks like it only reuses the Domain Adversarial Network once more.

**Questions:**

There was a great deal of confusion while reading the author's paper and the related literature mentioned:
1. whether the authors are the first to try to solve the work on Single-source cross-dataset ReID and why there are no other methods to solve this task in the compared approaches.
2. in the experimental comparisons, the data-enhanced method Random Erasure performs second best, while methods similar to the authors' perform poorly. This is confusing. Random erasure is not designed to work across datasets, methods such as ERM are. Can the authors give some analysis.

---

> ### Author Response · Authors · 2023-11-22
>
> Weakness 1: The performance of all methods is relatively poor since we use ResNet18 as the backbone for all methods. In the revised version, we will use stronger backbones such as ResNet50 or ViT.
>
> Weakness 2: The Class Adversarial Framework is similar to the Classical-DAN framework. However, we adopted it to the Re-ID problem with an extra loss (triplet loss) and a dynamic impact factor. Also, in contrast to DAN, which aims to learn domain-invariant representations, CAN aims to make the model robust to class space changes.
>
> Question 1: We are not the first paper to address the single-source cross-dataset ReID problem. In the revised version, we will explicitly point out the papers included in the related work specifically designed for Single-Source Cross-Dataset ReID.
>
> Question 2: In the revised version, we will analyse why classic methods perform well in certain conditions.

---

> > ### Comment · Reviewer_hNEm · 2023-11-22
> > **Lack of innovation is the main reason.**
> >
> > Adding more experiments is great. However, as with other reviewers, innovativeness is questioned. It is recommended that the authors make major improvements in their methodology.